# THE MANIFOLD ASSUMPTION AND DEFENSES AGAINST ADVERSARIAL PERTURBATIONS

## ABSTRACT

In the adversarial-perturbation problem of neural networks, an adversary starts with a neural network model $F$ and a point $\mathbf{x}$ that $F$ classifies correctly, and applies a *small perturbation* to $\mathbf{x}$ to produce another point $\mathbf{x}'$ that $F$ classifies *incorrectly*. In this paper, we propose taking into account *the inherent confidence information* produced by models when studying adversarial perturbations, where a natural measure of "confidence" is $\|F(\mathbf{x})\|_\infty$ (i.e. how confident $F$ is about its prediction?). Motivated by a thought experiment based on the manifold assumption, we propose a "goodness property" of models which states that *confident regions of a good model should be well separated*. We give formalizations of this property and examine existing robust training objectives in view of them. Interestingly, we find that a recent objective by Madry et al. encourages training a model that satisfies well our formal version of the goodness property, but has a weak control of points that are wrong but with low confidence. However, if Madry et al.'s model is indeed a good solution to their objective, then good and bad points are now distinguishable and we can try to embed uncertain points back to the closest confident region to get (hopefully) correct predictions. We thus propose embedding objectives and algorithms, and perform an empirical study using this method. Our experimental results are encouraging: Madry et al.'s model wrapped with our embedding procedure achieves almost perfect success rate in defending against attacks that the base model fails on, while retaining good generalization behavior.

## 1 INTRODUCTION

In the adversarial perturbation problem of neural networks, an adversary starts with a neural network model $F$ and a point $\mathbf{x}$ that $F$ classifies correctly (we assume that $F$ ends with a softmax layer, which is common in the literature), and applies a small perturbation to $\mathbf{x}$ to produce another point $\mathbf{x}'$ that $F$ classifies *incorrectly*. Szegedy et al. (2013) first noticed the vulnerability of existing (deep) neural networks to adversarial perturbations, which is a somewhat surprising phenomenon given the great generalization capability of these networks. Since then, a line of research (see, for example, Goodfellow et al. (2014); Papernot et al. (2016b); Miyato et al. (2017); Madry et al. (2017)) has been devoted to harden neural networks against adversarial perturbation. However, while modest progress has been made, until now there is still a large gap in successfully defending against more advanced attacks, such as attack by Carlini & Wagner (2017).

In this paper, we propose taking into account *the inherent confidence information produced by models* when studying adversarial perturbations. To this end, a natural measure of confidence is $\|F(\mathbf{x})\|_\infty$ (i.e., how confident $F$ is about its prediction?) We motivate this consideration with a thought experiment based on the manifold assumption (Zhu & Goldberg (2009); man) that is commonly made in unsupervised and semi-supervised learning, which states that *natural data points lie on (or near to) separate low dimensional manifolds for different classes.* Essentially, if one believes that deep neural networks learn to approximate well these low dimensional manifolds, then an ideal model should have the property that it can *confidently* distinguish points from natural manifolds as they are well separated due to the assumption. Moreover, since the learner never sees points that are far away from the natural manifolds, an ideal model should not claim confidence there.

Taking this perspective, we propose a *goodness property* which states that *confident regions of a good model should be well separated*. We give formalizations of this property and examine existing

robust training objectives in view of them. Interestingly, we find that a recent objective function by Madry et al. (2017) encourages training a model that satisfies well our formal version of the goodness property, in the sense that high-confidence predictions of different classes are well separated. On the other hand, our analysis also indicates that there could be many points with wrong but *low-confidence* predictions. Therefore, if Madry et al.'s model is indeed a good solution to their objective, then we can distinguish between good and bad points with confidence, and try to embed a low-confidence point back to confident regions to get (hopefully) the correct prediction.

We propose two embedding objectives: (1) $\delta$-Most Confident Neighbor ($\text{MCN}_\delta$), where $\text{MCN}_\delta(F, \mathbf{x}) = \arg\max_{\mathbf{z} \in N(x,\delta)} \|F(\mathbf{z})\|_\infty$ for a point $\mathbf{x}$, a radius parameter $\delta > 0$, and $N(x,\delta)$ the $\delta$-neighborhood around $x$. (2) $p$-Nearest Confident Neighbor ($\text{NCN}_p$), where $\text{NCN}_p(F, \mathbf{x}) = \arg\min_{\mathbf{z}} \|z - x\|$ subject to $\|F(\mathbf{z})\|_\infty \geq p$, for a point $\mathbf{x}$ and a confidence parameter $p \in (0, 1)$. With these, the end to end predictions become $F(\text{MCN}_\delta(F, x))$ and $F(\text{NCN}_p(F, x))$. We note that these objectives are *semantic*: They fail only when the model has a *confident but wrong* prediction in the neighborhood.

We perform an empirical study over CIFAR10. We first empirically validate that Madry et al.'s model is better, in view of our goodness property, than models trained without a robustness objective. We then give end-to-end defense results based on our method. Specifically, we propose using gradient based optimization, such as Carlini-Wagner attacks Carlini & Wagner (2017) (which, however, are now used for defense) to solve $\text{MCN}_\delta$ or $\text{NCN}_p$. Our empirical results are encouraging: (1) It achieves almost perfect success rate in defending against attacks that the base model fails on, and (2) It also retains the good generalization behavior of the base model.

The rest of the paper is organized as follows: We first discuss important prior work in Section 2 and some preliminaries in Section 3. Then Section 4 proposes the goodness property, and examines Madry et al.'s robust training objective function in view of the property. We then present embedding objectives and algorithms for handling low-confidence points in Section 5. Section 6 performs an empirical study where we validate that Madry et al.'s robust model satisfies well our goodness property, and then give defense results for our technique. Section 7 concludes with discussions on implications of our method.

## 2 PRIOR WORK

Szegedy et al. (2013) first observed the susceptibility of deep neural networks to adversarial perturbations. Since then, a large body of work have been devoted to studying hardening neural networks for this problem (a subset of work in this direction is Goodfellow et al. (2014); Papernot et al. (2016b); Miyato et al. (2017)). Simultaneously, another line of work have been devoted to devise more effective or efficient attacks (a small set of work in this direction is Moosavi-Dezfooli et al. (2016); Papernot et al. (2016a); Carlini & Wagner (2017)). Unfortunately, there still seems to be a large gap for the defense methods to defend against more sophisticated attacks, such as Carlini-Wagner attacks Carlini & Wagner (2017). For example, while the recent robust residual network constructed by Madry et al. (2017) achieves encouraging robustness results on MNIST, on CIFAR10 the accuracy against a strong adversary, such as attacks by Carlini & Wagner (2017), can be as low as $45.8\%$.

A dominant defense approach is to add to the training objective function an appropriate robustness component to improve robustness of the resulting model. Our method is fundamentally different and tries to complement training-based method by exploiting fundamental structures a model has learned. Our method has some remote similarities with a recent proposal by Lu et al. (2017), which tries to exploit features deep in the network (that is "semantics" of a model) to *detect* adversarial examples. However, beyond detecting adversarial attacks, our work takes a step further to compute correct prediction on the adversarial examples.

Our work is inspired by a recent line of research which demonstrates that neural networks are effective devices to represent low dimensional manifold data (for example, Rifai et al. (2011); Shaham et al. (2015); Basri & Jacobs (2016)). Specifically, in Basri & Jacobs (2016), the authors studied a specific type of manifolds called monotonic chains. They showed that neural networks can effectively represent data that both fall on, and near the manifolds (see "**Section 4. Error Analysis**" of the corresponding paper). While their network is not trained but is specifically constructed to fit the data, we believe that it has provided evidence that deep networks can extract intrinsic geometric

structures. To this end, one may interpret our work as arguing that if manifold assumption indeed holds, and the model approximates natural manifolds well, then it may not be surprising to see the coexistence of good generalization and adversarial perturbations. In particular, one may expect to see many wrong predictions though with low confidence.

## 3 PRELIMINARIES

As in existing work, such as Carlini & Wagner (2017); Papernot et al. (2016b), we define $F$ to be a neural network after the softmax layer. With this notation, the final classification is then $C_F(\mathbf{x}) = \arg\max_i F(\mathbf{x})_i$ where $F(\mathbf{x})_i$ gives the confidence of the network in classifying $\mathbf{x}$ for class $i$. We use $Z(\mathbf{x})$ to denote part of $F$ except the softmax layer. That is, $Z(\mathbf{x})$ computes the *logits* to be fed into the softmax function. We use $\mathcal{C}$ to denote the class of all labels. We need the following definitions:

**Definition 1** ($p$-confident point and $p$-confident region)**.** *Let $p \in [0,1]$ be a parameter, $F$ be a model, and $l \in \mathcal{C}$. A point $x$ is $p$-confident for label $l$ if $F(x)_l \geq p$. Further, the $p$-confident region of $F$ for label $l$ is $\mathcal{M}_l^p = \{\mathbf{x} \mid F(\mathbf{x})_l \geq p\}$. That is $\mathcal{M}_l$ is the set of points where $F$ has confidence at least $p$ for label $l$.*

**Definition 2.** *Let $d(\cdot, \cdot)$ be a distance metric between points, and $\mathcal{M}_1, \mathcal{M}_2$ be two sets of points, the distance between $\mathcal{M}_1$ and $\mathcal{M}_2$, $d(\mathcal{M}_1, \mathcal{M}_2)$ is defined to be $d(\mathcal{M}_1, \mathcal{M}_2) = \inf_{x \in \mathcal{M}_1, y \in \mathcal{M}_2} d(x, y)$.*

For example, a common distance metric used in studying adversarial perturbations is $d(x, y) = \|x - y\|_\infty$ where $\|\cdot\|_\infty$ is the infinity norm of a vector. Therefore $d(\mathcal{M}_1, \mathcal{M}_2)$ is the infimum of $\|x - y\|_\infty$ for $x \in \mathcal{M}_1, y \in \mathcal{M}_2$.

**Definition 3** (Cross entropy)**.** *Let $p, q$ be two discrete probability distributions, the* cross entropy *between $p, q$ is $\mathrm{H}(p, q) = -\sum_i p_i \log q_i$, where the log is base $2$.*

## 4 GOODNESS PROPERTY AND MADRY ET AL.'S OBJECTIVE

This section develops our main arguments. In particular:

- We propose a goodness property of models which states that *confident regions of a good model should be well separated.*
- We examine existing robust training objectives and demonstrate that an objective function of Madry et al. (2017) encourages very good separation for $p$-confident points with large $p$, but has a weak control of points that are wrong but with low confidence.

**Goodness Property**. Manifold assumption in unsupervised and semi-supervised learning states that natural data points lie on (or near to) separate low dimensional manifolds for different classes. Under this assumption what would an ideal model look like? Clearly, we would expect that an ideal model can *confidently* classify points from the manifolds, while *not* claiming confidence for points that are far away from those manifold. Therefore, we propose the following goodness property

*Confident regions of a good model should be well separated.*

The following definition captures well-separatedness in a strict sense,

**Definition 4** ($(p, \delta)$-separation)**.** *Let $p \in [0,1]$, $\delta \geq 0$, and $d(\cdot, \cdot)$ be a distance metric. $F$ is said to have $(p, \delta)$-separation if*

$$\inf_{l \neq l'} d(\mathcal{M}_l^p, \mathcal{M}_{l'}^p) \geq \delta$$

*where $\mathcal{M}_l^p$ and $\mathcal{M}_{l'}^p$ are $p$-confident regions for labels $l, l'$, respectively.*

It is also natural to consider the following probabilistic version of Definition 4,

**Definition 5** ($(p, q, \delta)$-separation)**.** *Let $\mathcal{D}$ be a data generating distribution, $p, q \in [0,1]$, $\delta \geq 0$, and $d(\cdot, \cdot)$ be a distance metric. $F$ is said to have $(p, q, \delta)$-separation if*

$$\Pr_{(x,y) \sim \mathcal{D}} \left[ (\exists y' \neq y, x' \in N(x, \delta)) \; F(x')_{y'} \geq p \right] \leq q,$$

*where $N(x, \delta) = \{x' \mid d(x, x') \leq \delta\}$.*

Note that these two definitions have some important differences: (1) Definition 4 is defined over sets of points while Definition 5 is defined point-wise. Further, Definition 5 allows certain points to be "bad" in the sense that they are not separated from confident points with wrong predictions. (2) Perhaps more importantly, Definition 5 depends on the data generating distribution, but Definition 4 is solely a property of the model. We are not aware of how to train a model to satisfy Definition 4. However, interestingly, we find that existing robust training can indeed give guarantees with respect to Definition 5. We next give an analysis of this.

**Objective function of Madry et al. (2017).** We now examine objective function used in Madry et al. (2017), and prove that it encourages training a model that satisfies Definition 5. Complementing this, we show that their objective has a weak control of points with low-confidence but wrong predictions. To start with, their objective function is of the form

$$\text{minimize} \quad \rho(\theta), \text{where } \rho(\theta) = \mathop{\mathbb{E}}_{(x,y)\sim\mathcal{D}} \left[ \max_{\Delta\in\mathcal{S}} L(\theta, x + \Delta, y) \right]. \tag{1}$$

where $\mathcal{D}$ is the data generating distribution, $\mathcal{S}$ is set of allowed perturbations (for example $\mathcal{S} = \{\Delta \mid \|\Delta\|_\infty \leq \delta\}$), and $L(\theta, x, y)$ is the loss of $\theta$ on $(x, y)$. Madry et al. (2017) uses cross entropy loss function:

$$L(\theta, x, y) = \mathrm{H}(\mathbf{1}_y, F_\theta(x)) = -\log F_\theta(x)_y$$

where $F_\theta$ is the model instantiated with parameters $\theta$, and $\mathbf{1}_y$ is the indicator vector of label $y$.

Denote $\kappa(\theta, x, y) = \max_{\Delta\in\mathcal{S}} L(\theta, x + \Delta, y)$, we have the following proposition,

**Proposition 1.** *If $\rho(\theta) \leq \varepsilon$, then*

$$\mathop{\Pr}_{(x,y)\sim\mathcal{D}} \left[ (\exists y' \neq y, x' \in x + \mathcal{S}) \, F_\theta(x')_{y'} \geq p \right] \leq -\frac{\varepsilon}{\log(1 - p)}$$

*That is, the probability of an $x' \in x + \mathcal{S}$ that is $p$-confident on a different label vanishes as $p \to 1$.*

*Proof.* Let $\mathcal{E}$ be the event $\{\exists y' \neq y, x' \in x + \mathcal{S}, F_\theta(x')_{y'} \geq p\}$. If $\mathcal{E}$ happens, then $F_\theta(x')_y \leq 1 - p$, and $\kappa(\theta, x, y) \geq -\log(1 - p)$, therefore by Markov's inequality[1],

$$\mathop{\Pr}_{(x,y)\sim\mathcal{D}}[\mathcal{E}] \leq \mathop{\Pr}_{(x,y)\sim\mathcal{D}} \left[ \kappa(\theta, x, y) \geq -\log(1 - p) \right] \leq -\frac{\varepsilon}{\log(1 - p)}.$$

The proof is complete. □

This immediately generates the following corollary,

**Corollary 1.** *Let $\mathcal{S}$ be a region defined as $\{\Delta \mid d(\Delta, \mathbf{0}) \leq \delta\}$. If $\rho(\theta) \leq \varepsilon$, then the model $F_\theta$ is $(p, -\frac{\varepsilon}{\log(1-p)}, \delta)$-separated.*

This indicates that even if a model is only a somewhat good solution to (1), meaning $\rho(\theta) \leq \varepsilon$ for somewhat small $\varepsilon$, (note that in reality $\mathcal{D}$ is unknown and a learner can only approximate (1)), then $p$-confident points will be well separated (in the sense of Definition 5) as soon as $p$ increases.

The above proposition considers a situation where we have a confident but wrong point (with confidence $p$). What about points that have wrong but low-confidence predictions? For example, the confidence on the wrong label is only $\frac{1}{2} + \nu$ for some tiny $\nu$? Note that by setting $p = 1/2$ we derive immediately a bound $\varepsilon$ (much weaker than $-\varepsilon/\log(1 - p)$ for large $p$) on the probability that bad events happen. Further, without further assumptions, this bound in tight so that in a worst-case sense that there exists a bad model so that bound $\varepsilon$ is achieved. We have the following proposition.

**Proposition 2.** *Let $F_\theta$ be a neural network parameterized by $\theta$, and $C_{F_\theta}$ be the classification network. If $\rho(\theta) \leq \varepsilon$, then*

$$\mathop{\Pr}_{(x,y)\sim\mathcal{D}} \left[ (\exists y' \neq y, x' \in x + \mathcal{S}) \, C_{F_\theta}(x') = y' \right] \leq \varepsilon.$$

*Further the bound is tight.*

---

[1] Let $X$ be a nonnegative random variable and $a > 0$, Markov's inequality says that $\Pr[X \geq a] \leq \mathbb{E}[X]/a$.

*Proof.* Let $\mathcal{E}$ be the event $\{(\exists y' \neq y, x' \in x + \mathcal{S}) \, C_{F_\theta}(x') = y'\}$. If $\mathcal{E}$ happens then $F_\theta(x')_y \leq \frac{1}{2}$ (otherwise $x'$ will be classified as $y$), and so $\kappa(\theta, x, y) \geq -\log(1/2) = 1$. On the other hand, if $\mathcal{E}$ does not happen, then we can lower bound $\kappa(\theta, x, y)$ by $0$. Therefore

$$\varepsilon \geq \mathbb{E}[\kappa(\theta, x, y)] \geq \Pr[\neg \mathcal{E}] \cdot 0 + \Pr[\mathcal{E}] \cdot 1 = \Pr[\mathcal{E}].$$

Tightness follows as we can force equality for each of the inequalities. The proof is complete. $\square$

Contrasting Proposition 1 and 2, we note the following:

- First, let us note that the objective function 1 is defined with respect to the data generating distribution $\mathcal{D}$, which is unknown, and in reality we only have a training set to approximate the objective. Also, cross entropy function is unbounded (so in particular it can happen that $\rho(\theta) \geq 1$). Therefore, one may only expect that $\rho(\theta)$ is somewhat small.

- Nevertheless, even with a somewhat large $\rho(\theta)$, Proposition 1 indicates that high-confidence predictions of different classes will be well separated as soon as confidence $p$ increases. On the other hand, Proposition 2 then indicates that there may still be many points that have *wrong* predictions with *low confidence*.

- Finally, if we take the manifold point of view again, then intuitively it seems harder to control low-confidence points than to separate confident regions. This is because natural manifolds reside in lower dimensions and so there can be a large volume of low-confidence points at the confidence boundary. This thus indicates that handling low-confidence points may be the core difficulty in getting a good solution to (1).

## 5 EMBEDDING

Our analysis from previous section shows that there can be good models with well-separated confident regions, yet there is essentially no control for uncertain points. However, since now uncertain points are distinguishable as they have low confidence, a natural plan to handle these points is to try to "embed them back" to confident regions. The rest of this section presents objective functions and algorithms for this purpose.

**Objective Functions.** We propose two objective functions for embedding.

**Definition 6** (**Most Confident Neighbor Objective**). *Let $F$ be a model, $\|\cdot\|$ be a norm, and $\delta > 0$ be a real number. The $\delta$-Most Confident Neighbor objective (or simply $\text{MCN}_\delta$ objective) computes:*

$$\text{MCN}_\delta(F, \mathbf{x}) \equiv \underset{\mathbf{z} \in N(\mathbf{x}, \delta)}{\arg\max} \|F(\mathbf{z})\|_\infty. \tag{2}$$

*where $N(\mathbf{x}, \delta)$ is the $\delta$-ball around $\mathbf{x}$ with respect to $\|\cdot\|$, and $\|\cdot\|_\infty$ denotes the $L_\infty$-norm.*

**Definition 7** (**Nearest Confident Neighbor Objective**). *Let $F$ be a model, $\|\cdot\|$ be a norm, and $p \in (0, 1)$ be a real number. The $p$-Nearest Confident Neighbor objective (or simply $\text{NCN}_p$ objective) computes:*

$$\text{NCN}_p(F, \mathbf{x}) \equiv \underset{\mathbf{z}}{\arg\min} \|\mathbf{z} - \mathbf{x}\| \text{ subject to } \|F(\mathbf{z})\|_\infty \geq p. \tag{3}$$

*where $\|\cdot\|_\infty$ denotes the $L_\infty$-norm.*

In short, MCN considers a $\delta$-neighborhood of a point and use the *most confident* point in the neighborhood for prediction. On the other hand, NCN considers the *nearest* neighbor that achieves certain confidence threshold.

**Model Shell.** From an algorithmic point of view, our approach essentially tries to wrap around a learned model with a "shell" to handle points that are "slightly" out of the confident regions it has learned. There are many possible shells for this purpose. For example, a valid such shell is weighted majority vote. We thus begin by giving some general definitions.

**Definition 8** (**Model Shell**). *Let $F$ be a model. A model shell $G$ is an algorithm that takes as its input $F$ and a point $\mathbf{x} \in \mathbb{R}^d$ to classify, and outputs a* softmax layer $G(F, \mathbf{x})$. *In this paper we only consider the case where $F$ is a model that is chosen apriori and fixed. In this case, the resulting model for classification is $G(F, \cdot)$, which maps a feature vector to a softmax layer. In this scenario, we say that $F$ is the* base model *of the model shell $G$, and denote $G(F, \cdot)$ as $G[F]$.*

Model shell captures the intuition that we want to wrap around an existing model with another algorithm that can exploit semantics of the base model. In this paper, we only examine model shells with a restricted structure, which we call *factor model shells*. We define *search factor*,

**Definition 9** (**Search Factor**). *A search factor $H$ is an algorithm that takes as its input a model $F$, and a point $\mathbf{x} \in \mathbb{R}^d$ to classify, and output another point $\mathbf{x}' \in \mathbb{R}^d$. If $F$ is chosen and fixed apriori we say that $F$ is the* base model *of the search factor $H$, and denote $H(F, \cdot)$ as $H[F]$.*

**Definition 10** (**Factor Model Shell**). *A model shell $G$ is said to be a* factor model shell *if $G$ can be written as a composition of $F \circ H$ where $H$ is a search factor. In other words, $G(F, \mathbf{x})$ works as*

$$G(F, \mathbf{x}) \equiv (F \circ H[F])(\mathbf{x}) = F(H(F, \mathbf{x})).$$

A factor model shell captures our intuitions from MCN or NCN objectives. That is, we first apply a search factor $H$ to find another feature point $\mathbf{x}' = H(F, \mathbf{x})$, and then apply the base model $F$ on $\mathbf{x}'$ to produce a final prediction. As a result, a factor model shell for $\text{MCN}_\delta$ objective should compute the final prediction as:

$$(F \circ \text{MCN}_\delta)(\mathbf{x}) = F(\text{MCN}_\delta(F, \mathbf{x})), \tag{4}$$

and accordingly for $\text{NCN}_p$ objective:

$$(F \circ \text{NCN}_p)(\mathbf{x}) = F(\text{NCN}_p(F, \mathbf{x})) \tag{5}$$

We note the following:

- Exact computations of (4) and (5) provide *semantic* defenses. That is, the final prediction is wrong only if the model has *a confident but wrong* prediction in the neighborhood. Furthermore, our discussion from Section 4 indicates that for good models, wrong predictions should also have *low confidence*. Therefore, embedding should be effective in fixing low-confidence errors of good models.

- We note that the resulting factor model shells, $(F \circ \text{MCN}_\delta)$ and $(F \circ \text{NCN}_p)$ are *non-differentiable* as $\text{MCN}_\delta(F, x)$ and $\text{NCN}_p(F, x)$ are non-differentiable with respect to $x$ (they are argmin or argmax over a ball). This renders existing attacks fail to work directly to attack $F \circ \text{MCN}_\delta$ or $F \circ \text{NCN}_p$, since they all require the assumption that *the output softmax layer is differentiable with respect to the input feature vector $x$*[2] As a result, this forces attacks to either attack the base model $F$ and then transfer attacks to the model shell, or has to find some differentiable approximations or derivative-free optimization to attack $\text{MCN}_\delta$ and $\text{NCN}_p$ directly.

**Algorithms.** We now give one concrete algorithms to solve (2) and (3). To start with, we note that the optimization for solving $\text{MCN}_\delta$ (2) is nothing but for each label $t \in \mathcal{C}$ we try to find

$$\mathbf{z}_t = \underset{\mathbf{z} \in N(\mathbf{x}, \delta)}{\arg\max} \, F(\mathbf{z})_t \tag{6}$$

---

[2] For example, let us examine one objective function used in Carlini-Wagner Carlini & Wagner (2017), which is one of the strongest attacks known. They use an objective function of the form ($f_5$ on pp. 6 of their paper Carlini & Wagner (2017)):

$$f(\mathbf{x}') = -\log(2F(\mathbf{x}')_t - 2)$$

where $F(\mathbf{x}')_t$ denotes the coordinate of $F(\mathbf{x}')$ at label $t$. This objective is differentiable in $\mathbf{x}'$ as $F$ is differentiable in $\mathbf{x}'$. We note that such a first-order assumption is natural for neural networks because neural networks are typically composition of differentiable functions. With first order information, attacks can "relatively accurately" follow the gradient direction to modify the image feature so as to increase the confidence of the model in *incorrect labels* and thus produce adversarial perturbations.

and then compute $\mathbf{z}_{t^*}$ for $t^* = \arg\max_t F(\mathbf{z}_t)_t$.

Now, however, we can solve (6) using any preferred gradient-based optimization (for example, projected gradient descent Nocedal & Wright (2006)). This thus gives the following factor model shell:

---

**Algorithm 1** Solving $\texttt{MCN}_\delta$ objective.

---

**Input:** $\mathbf{x}$ a feature vector, $\delta > 0$ a real parameter, a base model $F$, any gradient-based optimization algorithm $\mathcal{O}$ to solve the constrained optimization problem defined in (6).
 1: **function** MCNOracleShell($\mathbf{x}, \delta, F$)
 2:     **for** $t \in \mathcal{C}$ **do**
 3:         $\mathbf{z}_t \leftarrow \mathcal{O}(\mathbf{x}, F, t)$
 4:     **return** $\mathbf{z}_{t^*}$ where $t^* = \arg\max_{t \in \mathcal{C}} F(\mathbf{z}_t)_t$

---

We note that solving (6) is similar to *a targeted adversarial attack which tries to modify* $\mathbf{x}$ *so as to increase the confidence on label* $t$. Therefore *an adversarial attack*, such as strong attack proposed in Carlini & Wagner (2017), can be used as $\mathcal{O}$ here.

Another concern is that gradient-based optimization may only find local optimal in the neighborhood. However, we note that global optimal may not be needed for our method to work, as long as local minima of the correct class are "separated" from local minima of incorrect classes. More specifically, let $l$ be the correct class, and

$$\mathcal{S}_l = \{z : z \in N(x, \delta), z \text{ is a local maximal of } \|F(x)\|_\infty, \|F(z)\|_\infty = F(z)_l\}$$

and

$$\mathcal{S}_{-l} = \{z : z \in N(x, \delta), z \text{ is a local maximal of } \|F(x)\|_\infty, \|F(z)\|_\infty \neq F(z)_l\}$$

then local optimal works as soon as $\inf_{z \in \mathcal{S}_l} \|F(z)\|_\infty > \sup_{y \in \mathcal{S}_{-l}} \|F(y)\|_\infty$.

We can also take a similar path to solve $\texttt{NCN}_p$ objective with standard numeric optimization. Note that $\|F(\mathbf{z})\|_\infty \geq p$ is equivalent to $\exists t, F(\mathbf{z})_t \geq p$. Therefore, we can solve

$$\mathbf{z}_t = \arg\min_{F(\mathbf{z})_t \geq p} \|\mathbf{z} - \mathbf{x}\| \tag{7}$$

for every $t$, and then compute $\mathbf{z}_{t^*}$ where $t^* = \arg\min_t \|\mathbf{z}_t - \mathbf{x}\|$. By a similar reasoning as above, $\mathbf{z}_{t^*}$ is the solution to problem 3. This thus gives algorithm 2 for $\texttt{NCN}_p$.

---

**Algorithm 2** Solving $\texttt{NCN}_p$ objective.

---

**Input:** $\mathbf{x}$ a feature vector, $p > 0$ a real parameter, a base model $F$, any gradient-based optimization algorithm $\mathcal{O}$ to solve the constrained optimization problem defined in (7).
 1: **function** NCNOracleShell($\mathbf{x}, p, F$)
 2:     **for** $t \in \mathcal{C}$ **do**
 3:         $\mathbf{z}_t \leftarrow \mathcal{O}(\mathbf{x}, F, t)$
 4:     **return** $\mathbf{z}_{t^*}$ where $t^* = \arg\min_t \|\mathbf{z}_t - \mathbf{x}\|$

---

Solving (7), however, is not trivial. A standard method is the augmented Lagrangian method Bertsekas (1996) which turns the problem into solving a series of unconstrained optimization. For example, consider the following quadratic penalty function $Q(\mathbf{z}, \alpha, t) = \|\mathbf{z} - \mathbf{x}\| + \frac{\alpha}{2}\left([F(\mathbf{z}) - p]^-\right)^2$, where $[y]^-$ denotes $\max\{y, 0\}$. Then we can solve the unconstrained optimization $\min Q(\mathbf{z}, \alpha, t)$ for a series of different $\alpha$, and terminates once we find a satisfying solution. The intuition is that (i) solving the unconstrained optimization is easy, (ii) by increasing the coefficient $\alpha$, we can force the minimizer of the penalty function closer to the feasible region $F(\mathbf{z}) \geq p$, and (iii) for well chosen $\alpha$, the minimizer of the penalty function is close enough to that of problem 7. Other classic constrained optimization algorithms to solve this problem are interior methods and sequential quadratic programming Nocedal & Wright (2006). We also note a recent work Bienstock & Michalka (2014) which specifically focuses on optimizing convex objective over non-convex constraints, into which problem 7 falls as well.

## 6 EMPIRICAL STUDY

In this section we perform an empirical study of our approach. Note that there are three basic components in a defense with our method: (1) the *base model* we choose, (2) the *embedding procedure* we choose to wrap around the base model, and (3) the *attack* we choose to evaluate the defense. Therefore we have the following key empirical questions:

1. *How well does a base model satisfy our goodness property?*
2. *How susceptible is the chosen base model to the chosen attack?* We want that the chosen attack is strong enough to break even a good base model on many points.
3. *How effective does our approach improve robustness of the base model?* Ideally, we want to see a significant improvement of robustness for a good base model.
4. *Will our approach change the generalization behavior of the base model?* Since embedding may change predictions, we need to justify if there is little or no change in generalization.

A summary of our findings are as follows:

- We empirically validate that the model from Madry et al. (2017) is significantly better than models without robustness training, according to our goodness property.
- We thus use Madry et al.'s model as the base model, and attack the model using attacks from Carlini & Wagner (2017) (denoted as CW attack). We use CIFAR10 Krizhevsky (2009) to evaluate the robustness and generalization. As reported by Madry et al., their model is still fairly susceptible to CW attack over CIFAR10, and the accuracy against CW attack can be as low as $45.8\%$, *which indicates a significant room for improving robustness*.
- We choose MCNOracleShell (Algorithm 1) and use again CW attack but now as embedding. The resulting model shell is called CarliniWagnerShell. We find that embedding *significantly improves the robustness of the base model* over 5 batches of points we evaluated, reducing the attack success rate from $30\%$ to $1.33\%$.
- Finally, *embedding incurs almost no effect on generalization.* In fact, there are only three such points that the embedding procedure changes the predictions from correct to wrong. We note that the model has very low confidence ($\sim 50\%$) on these points, and thus are difficult to defend as predicted by our theoretical analysis.

**Limitations of our experiments.** Due to the complexity of the Madry et al.'s residual network model, our embedding procedure takes a long time to compute and we have only tested on 150 points which the base model classifies correctly (we do not evaluate robustness for test points that have incorrect predictions in the first place). While our current experiments are limited, they corroborate well our theoretical analysis that embedding should work well with a good model to resolve adversarial perturbation problem. We have made our model and code publicly available cod, in the hope of triggering more thoughts and feedback.

**Experimental Setup** We use CIFAR10 Krizhevsky (2009) which consists of 60000 color images of objects where 50000 are for training and 10000 are for testing. Each image is of size $32 \times 32$ pixels, and there are ten labels $\{0, 1, \ldots, 9\}$ for classification. We use Carlini-Wager attacks (CW attack) for two purposes: (i) *as an attack to challenge the defense.* In this case, we use the original attacks to make attacks as strong as possible. We apply *non-targeted* attacks (namely changing classification is the purpose) with $L_\infty$ norm (i.e. try to find a perturbation of minimal $L_\infty$ norm). To set the norm bound $\delta$, we note that Madry et al. (2017) uses a norm bound $\delta = 8.0$. Since we normalize images when applying the CW attack, we set $\delta = \frac{8}{255}$. A perturbation is valid if and only if it both changes the classification and its $L_\infty$ norm is at most $\delta$. (ii) *as an embedding procedure for instantiating* MCNOracleShell. In our actual CarliniWagnerShell implementation, we slightly tweak parameters of CW attack so that the CW attack used as embedding is weaker but faster than the one used for attacking. This makes our results *stronger* because only the defense is weaker.

**Experimental Methodology** We perform two experiments. First, we compare two models in view of the goodness property: (i) The *robust model* trained by Madry et al. (2017), and (ii) a *natural model* that has the same model architecture, but is trained without a robust objective. The comparison method is based on Proposition 1. Let $\mathcal{P} = \{p \mid p \in [0.5, 1]\}$ be a set of probability values. We

first sample a batch of points $K$. For each $p \in \mathcal{P}$, we try to find, for each point $(x, y)$ in $K$, a point that is predicted as $y' \neq y$ with confidence $\geq p$ in its $\delta$-neighborhood. For each $p$ we get a fraction $s_p$ of points in $K$ that such confident attacks can be found. We find that *the robust model has much smaller $s_p$ for larger $p$ and hence it satisfies our the goodness property better.*

We then evaluate our end-to-end defense. We go through a random permutation of test data points, and measure the following for a point that the base model predicts correctly: (1) *Susceptibility*. That is, whether CW attack can successfully attack this point. (2) *Robustness*. There are two cases: (i) the model is already robust to CW attack where we expect CarlinWagnerShell retains the robustness, and (ii) the model is vulnerable to CW attack, where we expect CarliniWagnerShell to harden the model. (3) *Generalization*. That is, whether CarliniWagnerShell still predicts correctly on the point.

**Comparing models using the goodness property.** From Definition 5 and Proposition 1, our goal is to statistically estimate $\Pr_{(x,y) \sim \mathcal{D}} \left[ (\exists y' \neq y, x' \in N(x, \delta)) F(x')_{y'} \geq p \right]$. As mentioned above, we compare the *robust model* in Madry et al. (2017) and its *natural variant*. Let $\mathcal{E}_b$ denote the bad event $(\exists y' \neq y, x' \in N(x, \delta)) F(x')_{y'} \geq p$. We randomly sample batches of data points from the test set, and compute frequency that $\mathcal{E}_b$ happens for each batch. To do so, we attack samples using a modified CW attack, which tries to generate an adversarial example within the norm bound where the model classifies incorrectly with high confidence. Table 1 summarizes the results.

| | # successful attacks | |
| --- | --- | --- |
| | Robust model | Natural model |
| Batch #1 | 2 | 26 |
| Batch #2 | 3 | 27 |
| Batch #3 | 4 | 26 |
| Batch #4 | 5 | 28 |
| Batch #5 | 2 | 29 |
| Batch #6 | 4 | 25 |
| Batch #7 | 0 | 28 |
| Batch #8 | 2 | 30 |
| Batch #9 | 3 | 28 |
| Batch #10 | 1 | 30 |
| Total | 26 | 277 |

Table 1: Results from testing the goodness property of base models. Each batch has 30 random samples. For each column, we record the number of samples that we can successfully find attacks of confidence at least $p = 0.9$ against the model corresponding to the column.

With these statistics we can thus estimate $(p, q, \delta)$-separation (Definition 5) of the models in comparison. We have (details are deferred to Appendix A).

**Proposition 3** (Separation from statistics). *The following two hold:*

- *With probability at least .9 the robust model in Madry et al. (2017) satisfies*

$$\Pr_{(x,y) \sim \mathcal{D}} \left[ (\exists y' \neq y, x' \in N(x, \delta)) F(x')_{y'} \geq .9 \right] \leq \frac{14}{75} = 0.18667\ldots$$

  *That is, the robust model has $(\frac{9}{10}, \frac{14}{75}, \frac{8}{255})$-separation.*

- *With probability at least .9 the natural model satisfies*

$$\Pr_{(x,y) \sim \mathcal{D}} \left[ (\exists y' \neq y, x' \in N(x, \delta)) F(x')_{y'} \geq .9 \right] \geq \frac{247}{300} = 0.82333\ldots$$

  *That is, the natural model does* not *have $(\frac{9}{10}, q, \frac{8}{255})$-separation for $q < \frac{247}{300}$.*

**End-to-end defense results.** We have evaluated 5 non-overlapping batches from the test set (each batch consists of 30 points, thus 150 points in total), where the base model classifies test points in

these batches correctly (in other words, we skip test points where the base model predicts incorrectly). We find that the base model is susceptible to CW attack for 56 of them (thus the attack success rate is 37.33%). We find the following after applying CarliniWagnerShell: First, the points that were robust with base model remain robust, thus we do not introduce new vulnerabilities. Second, among the 56 vulnerable points of base model, for 53 of them CarliniWagnerShell now is both correct and *robust* against attacks on which the base model fails. Finally, for the remaining 3 points: For only one of them, CarliniWagner shell produces wrong prediction but is robust (makes consistent predictions for both vanilla the perturbed points), and for the remaining two CarliniWagnerShell is both incorrect and not robust. As a summary, the attack success rate reduces from 37.33% to $2/150 = 1.33\%$, and the accuracy retained is $147/150 = 98\%$. The results are summarized in Table 2.

|  | Base model | CarliniWagnerShell-equipped model | |
|---|---|---|---|
|  | Attack success rate | Attack success rate | Retained accuracy |
| Batch #1 | 30 % | 0 % | 96.67 % |
| Batch #2 | 33.33 % | 0 % | 100 % |
| Batch #3 | 40 % | 0 % | 100 % |
| Batch #4 | 33.33 % | 0.07 % | 99.33 % |
| Batch #5 | 46.67 % | 0 % | 100 % |
| Average | 37.33 % | 1.33 % | 98 % |

Table 2: Summary of the experimental results. Among the 150 test points where the base model is correct. For 56 of them the base model is vulnerable to CW attacks. Applying CarliniWagnerShell does not introduce new vulnerable points. And for 53 of the 56 vulnerable points, CarliniWagnerShell is both correct and robust against the attacks that are successful on the base model. For one of the remaining three points, CarliniWagnerShell is incorrect but robust. For the remaining two CarliniWagnerShell is both incorrect and not robust.

**Wrong data points, confidence, and the manifold boundary.** We now describe the three data points where CarliniWagnerShell fails. We observe that for these three points the model has low confidence, and so by our theory it is expected to be hard to defend. Taking again a manifold point of view, we suspect that one possibility why these points have low confidence is that "natural manifolds" (which are *spaces of points generated by some computational devices*) are not really well separated. In the following we examine each of these three points.

From the first batch, we have a single point that CarliniWagnerShell gives a wrong prediction, but does not change its prediction after the image is adversarially perturbed (and thus is robust, but not correct). The image is presented in Figure 1(a). We note that the model gives low confidence ($\sim 50\%$) on the original image. (where the correct label is a "cat"), and it has significantly higher confidence ($\sim 75\%$) on the perturbed image (classified as a "frog"). In this case, Proposition 1 only indicates very weak separation, (indeed, $\varepsilon$) and so it is expected to be hard to defend. Figure 1 depicts the images, with a similar frog image for comparison. We note that these points as generated by some cameras have essentially ignored the "size" information of different objects (can we tell from these images that a cat should be bigger than a frog?), and thus may not be well separated.

From the fourth batch, we have two points that CarlniWagnerShell gives us wrong predictions, and changes its predictions after the images are adversarially perturbed. Figure 2(a) and 2(f) depict these images and corresponding results. The situation is similar to the first image above where model predictions have low confidence: the base model gives low confidence on the original image ($\sim 50\%$ for "automobile" 2(a), and $\sim 60\%$ for "airplane" 2(f)) and adversarially perturbed images ($\sim 60\%$ for 2(a), and $\sim 50\%$ for 2(f). Both predictions are "ship"). For each row we attach an image which is a valid "ship" image but resembles "automobile" and "airplane" images in shape. Again, we suspect that the underlying natural manifolds are not well separated as one might expect.

# 7 DISCUSSION

We now discuss our method and results.

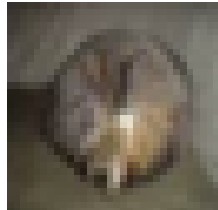 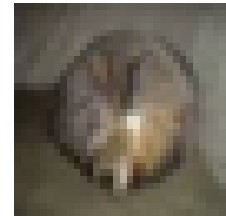 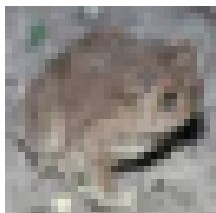

(a) Original cat image with unconfident prediction "cat"

(b) Adversarially perturbed image with a wrong and unconfident prediction "frog"

(c) A frog image

Figure 1: The point (a) where our CarliniWagnerShell makes a wrong prediction, and is consistent on the original test point and the perturbed one. We show the original image (a), CarliniWagnerShell perturbation of the image (b), and for comparison, a frog image (c) that the model is confident about.

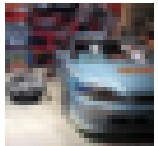 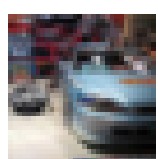 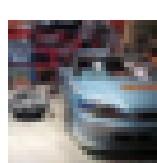 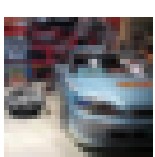 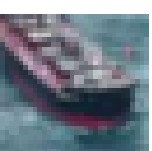

(a) Base model on $\mathbf{x}$, unconfident prediction "automobile"

(b) CarliniWagnerShell on $\mathbf{x}$, unconfident prediction "ship"

(c) Base model on $\mathbf{x}'$, unconfident prediction "ship"

(d) CarliniWagnerShell on $\mathbf{x}'$, unconfident prediction "automobile"

(e) A valid "ship" that resembles the "automobile"

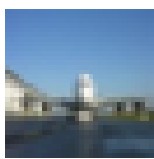 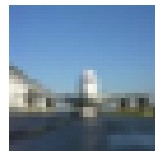 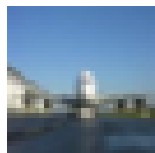 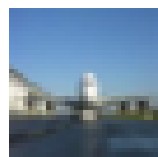 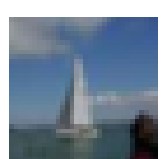

(f) Base model on $\mathbf{x}$, unconfident prediction "airplane"

(g) CarliniWagnerShell on $\mathbf{x}$, unconfident prediction "ship"

(h) Base model on $\mathbf{x}'$, unconfident prediction "ship"

(i) CarliniWagnerShell on $\mathbf{x}'$, unconfident prediction "airplane"

(j) A valid "ship" image that resembles the "airplane"

Figure 2: The images (a) and (e) are the points where our CarliniWagnerShell makes a wrong prediction. We show the original image on the first column, CarliniWagnerShell perturbations of the original images on the second column, adversarial perturbed images on the third column, and CarliniWagnerShell perturbations of the adversarial examples on the fourth column. Example ship images are presented on the last row for comparison.

**Our technical contributions.** Our first technical contribution is the proposal to take into account the inherent confidence information produced by models when studying adversarial perturbations. Our second technical contribution is a formulation of a goodness property of models, and an analysis of existing models in view of the property. Interestingly, and somewhat surprisingly, we find that a recent robust training objective function by Madry et al. (2017) encourages good separation of high-confidence points, but has essentially no control of low-confidence points. Our third contribution is the proposal of embedding to handle low-confidence points. Our final contribution is a first empirical study that validates Madry et al.'s model in terms of the goodness property, and further demonstrates that a good model, when wrapped with embedding, simultaneously achieves good generalization and almost perfect robustness.

**Interpretations of our results.** One interpretation of our results is that adversarial perturbations can *naturally coexist with good generalization*. While this is manifested in our analysis of Madry et al.'s

objective function, we think that this phenomenon naturally and generally exists if one takes again a manifold point of view: Since natural manifolds reside in low dimensions, it seems much more challenging to control the confidence boundary, where "adversarial perturbations" exist in abundance, than controlling separation of the learned structures (i.e. confident regions that approximate the underlying manifolds).

On the other hand, if all one cares about is robustness (decision does not change in a small neighborhood), then adversarial perturbation problem can be resolved by combining goodness property with embedding. For example, consider the following "good model:" If a data point is from the training set, then it outputs the correct label with confidence 1, otherwise it outputs a uniform distribution over labels. In other words, this model learns nothing but fitting the training set. We note that in this case, the adversarial perturbation problem is *only well defined* around the training points. Moreover, embedding now becomes 1-nearest neighbor search among the training points. As a result, the model is still *perfectly robust* with our method if training points are well separated.

**Highly confident predictions on random noises.** We note that several work shows that neural networks can have highly confident predictions on random noises (e.g., Nguyen et al. (2015)). In view of our work it is somewhat not surprising that neural networks can have such behaviors: These points are essentially ones that are *far away from* the "universe" the learner is asked to learn, and so if we do not control the training of neural networks to not claim confidence over the structure it has never seen, then it is valid to fit a model that has good behaviors on natural manifolds but also divergent behaviors outside. After all, why is a network supposed to work on points that are far away from the underlying natural manifolds, which is essentially the *data generating distribution?* Finally, we note that the adversarial perturbation problem is not well defined even near those points.

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

## A  BOUNDING THE PROBABILITY FOR $(p, q, \delta)$-SEPARATION

This section gives details of our estimation of $(p, q, \delta)$-separation from statistics in Table 1. Note that event $\mathcal{E}_b$ corresponds to a Bernoulli trial. Let $X_1, \ldots, X_t$ be independent indicator random variables, where

$$X_i = \begin{cases} 1 & \text{if } \mathcal{E}_b \text{ happens,} \\ 0 & \text{otherwise} \end{cases},$$

and $X = (\sum_{i=1}^{t} X_i)/t$. Recall Chebyshev's inequality:

**Theorem 1** (Chebyshev's Inequality). *For independent random variables $X_1, \ldots, X_t$ bounded in $[0, 1]$, and $X = (\sum_{i=1}^{t} X_i)/t$, we have $\Pr[|X - \mathbb{E}[X]| \geq \varepsilon] \leq \frac{\text{Var}[X]}{\varepsilon^2}$.*

In our case, $\mathbb{E}[X] = \mathbb{E}[X_1] = \cdots = \mathbb{E}[X_t]$ and let it be $\mu$, and let the computed frequency be $\hat{\mu}$ (observed value). Thus $\Pr[|\hat{\mu} - \mu| \geq \varepsilon] \leq 1/(4\varepsilon^2 t)$ since $\text{Var}[X] = \mu(1 - \mu)/t < 1/4t$. We thus have the following proposition about $(p, q, \delta)$-separation.

**Proposition 4.** *Let $\alpha, \varepsilon \in [0, 1]$. For sufficiently large $t$ where $\frac{1}{4\varepsilon^2 t} \leq 1 - \alpha$ holds, we have:*

- *(Upper bound) With probability at least $\alpha$, $\mu$ is smaller than $\hat{\mu} + \varepsilon$.*

- *(Lower bound) With probability at least $\alpha$, $\mu$ is bigger than $\hat{\mu} - \varepsilon$.*

For example, we have guarantees for $\alpha = .9$ by putting $\varepsilon = .1$ and $t \geq 250$.

