# OpenReview forum: "The Manifold Assumption and Defenses Against Adversarial Perturbations"
_ICLR.cc/2018/Conference — Reject_

### Official Review · AnonReviewer2 · 2017-11-21
**Not convincing.**

**Rating:** 3
**Confidence:** 3

**Review:**

The authors argue that "good" classifiers naturally represent the classes in a classification as well-separated manifolds, and that adversarial examples are low-confidence examples lying near to one of these manifolds. The authors suggest "fixing" adversarial examples by projecting them back to the manifold, essentially by finding a point near the adversarial example that has high confidence.

There are numerous issues here, which taken together, make the whole story pretty unconvincing.

The term "manifold" is used very sloppily. To be fair, this is unfortunately common in modern machine learning. An actual manifold is a specific mathematical structure with specific properties. In ML, what is generally hypothesized is that the data (often per class) lives "near" to some "low-dimensional" structure. In this paper, even the low-dimensionality isn't used --- the "manifold assumption" is used as a stand-in for "the regions associated with different classes are well-separated." (This is partially discussed in Section 6, where the authors point out correctly that the same defense as used here could be used with a 1-nn model.) This is fine as far as it goes, but the paper refs Basri & Jacobs 2016 multiple times as if it says anything relevant about this paper: Basri & Jacobs is specifically about the ability of deep nets to fit data that falls on (actual, mathematical) manifolds. This reference doesn't add much to the present story.

The essential argument of the paper rests on the "Postulate: (A good model) F is confident on natural points drawn from the manifolds, but has low confidence on points outside of the manifolds."

This postulate is sloppy and speculative. For instance, taken in its strong form, if believe the postulate, then a good model:
1. Can classify all "natural points" from all classes with 100% accuracy.
2. Can detect adversarial points with 100% accuracy because all high-confidence points are correct classifications and all low-confidence points are adversarial.
3. All adversarial examples will be low-confidence.

Point 1 makes it clear that no good model F fully satisfying the postulate exists --- models never achieve 100% accuracy on difficult real-world distributions. But the method for dealing with adversarial examples seems to require Points 2 and 3 being true.

To be fair, the paper more-or-less admits that how true these points are is not known and is important. Nevertheless, I think this paper comes pretty close to arguing something that I *think* is not true, and doesn't do much to back up its argument. Because of the quality of the writing (generally sloppy), it's hard to tell, but I believe the authors are basically arguing that:
a. You can generally easily detect adversarial points because they are low confidence.
b. If you go through a procedure to find a point near your adversarial point that is high-confidence, you'll get the "correct" (or perhaps "original") class back.

I think b follows from a, but a is extremely suspect. I do not personally work in adversarial examples, and briefly looking at the literature, it seems that most authors *do* focus on how something is classified and not its confidence, but I don't think it's *that* hard to generate high-confidence adversarial examples. Early work by Goodfellow et al. ("Explaining and Harnessing Adversarial Examples", Figure 1, shows an example where the incorrect classification has very high confidence. The present paper only uses Carlini-Wagner attacks. From a read of Carlini-Wagner, it seems they are heavily concerned with finding *minimal* perturbations to achieve a given misclassification; this will of course produce low-confidence adversaries, but I see no reason why this is a general property of all adversarial examples.

The experiments are weak. I applaud the authors for mentioning the experiments are very preliminary, but that doesn't make them any less weak.

What are we to make of the one image discussed at the end of Section 5 and shown in Figure 1? The authors note that the original image gives low-confidence for the correct class. (Does this mean that the classifier isn't "good"? Is it evidence against some kind of manifold assumption?) The authors note the adversarial category has significantly higher confidence, and say "in this case, it seems that it is the vagueness of the signals/data that lead to a natural difficulty." But the signals and data are ALWAYS vague. If they weren't, machine learning would be easy. This paper proposes something, looks at a tiny number of examples, and already finds a counterexample to the theory. What's the evidence *for* the theory?

A lot of writing is given over to how this method is "semantic", and I just don't buy it. The connection to manifolds is weak. The basic argument here is really "(1) If our classifiers produce smooth well-separated high-confidence regions, (2) then we can detect adversaries because they're low-confidence, and (3) we can correct adversaries by projecting them back to high-confidence." (1) seems vastly unlikely to me based on all my experience: neural nets often get things wrong, they often get things wrong with high confidence, and when they're right, the confidence is at least sometimes low. The authors use a sloppy postulate about good models and so could perhaps argue I've never seen a good model, but the methods of this paper require a good model. (2) seems to follow logically from (1). (3) is also suspect --- perturbations which are *minimal* can be corrected as this paper does (and Carlini-Wagner attacks are minimal by design), but there's no reason to expect general perturbations to be minimal.

The writing is poor throughout. It's generally readable, but the wordings are often odd, and sometimes so odd it's hard to tell what was meant. For instance, I spent awhile trying to decide whether the authors assumed common classifiers are "good" (according to the postulate) or whether this paper was about a way to *make* classifiers good (I eventually decided the former).

---

### Official Review · AnonReviewer1 · 2017-11-25
**Interesting ideas but at the current stage, this seems to be a preliminary work that is not well matured yet!**

**Rating:** 4
**Confidence:** 3

**Review:**

The manuscript proposes two objective functions based on the manifold assumption as defense mechanisms against adversarial examples. The two objective functions are based on assigning low confidence values to points that are near or off the underlying (learned) data manifold while assigning high confidence values to points lying on the data manifold. In particular, for an adversarial example that is distinguishable from the points on the manifold and assigned a low confidence by the model, is projected back onto the designated manifold such that the model assigns it a high confidence value. The authors claim that the two objective functions proposed in this manuscript provide such a projection onto the desired manifold and assign high confidence for these adversarial points. These mechanisms, together with the so-called shell wrapper around the model (a deep learning model in this case) will provide the desired defense mechanism against adversarial examples.

The manuscript at the current stage seems to be a preliminary work that is not well matured yet. The manuscript is overly verbose and the arguments seem to be weak and not fully developed yet. More importantly, the experiments are very preliminary and there is much more room to deliver more comprehensive and compelling experiments.

---

### Official Review · AnonReviewer4 · 2017-11-27

**Rating:** 5
**Confidence:** 3

**Review:**


1) Summary
This paper proposes a new approach to defending against adversarial attacks based on the manifold assumption of natural data. Specifically, this method takes inputs (possibly coming from an adversarial attack), project their semantic representation into the closest data class manifold. The authors show that adversarial attack techniques can be with their algorithm for attack prevention. In experiments, they show that using their method on top of a base model achieves perfect success rate on attacks that the base model is vulnerable to while retaining generalizability.


2) Pros:
+ Novel/interesting way of defending against adversarial attacks by taking advantage of the manifold assumption.
+ Well stated formulation and intuition.
+ Experiments validate the claim, and insightful discussion about the limitations and advantages of the proposed method.

3) Cons:
Number of test examples used too small:
As mentioned in the paper, the number of testing points is a weakness. There needs to be more test examples to make a strong conclusion about the method’s performance in the experiments.

Comparison against other baselines:
Even though the method proposes a new approach for dealing with adversarial attacks using Madry et al. as base model, it would be useful to the community to see how this method works with other base models.

Algorithm generalizability:
As mentioned by the authors, their method depends on assumptions of the learned embeddings by the model being used. This makes the method less attractive for people that may be interested in dealing with adversarial examples in, for example, reinforcement learning problems. Can the authors comment on this?

Additional comments:
The writing needs to be polished.


4) Conclusion:
Overall, this is a very interesting work on how to deal with adversarial attacks in deep learning, while at the same time, it shows encouraging results of the application of the proposed method. The experimental section could improve a little bit in terms of baselines and test examples as previously mentioned, and also the authors may give some comments on if there is a simple way to make their algorithm not depend on assumptions of the learned embeddings.

---

### Author Response · Authors · 2017-12-13
**First responses to most important concerns and first revision**

We appreciate the insightful comments. Our ideas seem controversial and have received very different opinions. Given the time constraint we want to give first responses for the most important concerns. We have also submitted a first revision, which gives theoretical justification of our ideas. We have also empirically validate that Madry et al.'s model is better.

1. What is this paper about?

In short, we want to propose taking into account the confidence information produced by models when studying adversarial perturbations. Manifold assumption is important for us to arrive at the conclusion that "A good model must have well-separated confident regions." We view this as a goodness property that gives a clue to adversarial perturbation problem. Specifically, this paper: (1) proposes a goodness property of a model, (2) argues that one may still have to handle low-confidence points with a good model, and (3) identifies a good model from literature and wraps it with embedding and evaluates the result.

2. What is claimed lacks backup, and is unlikely to hold for neural networks.

Interestingly, we can now prove that Madry et al.'s objective encourages a good model. Specifically, Section 4 of the revision gives formalizations of the goodness property and analyzes Madry et al.'s objective. We prove: (1) Even with a somewhat good solution to the objective, high-confidence points with different predictions will be well separated soon as confidence increases. (2) Controlling low-confidence points with wrong predictions is the core challenge to a good solution to the objective. These results are a bit unexpected, and corroborate our intuitions derived from the manifold assumption.

We are also empirically validating that Madry et al.'s robust model is good: Table 1 of the revision compares two models that share the same architecture where one is trained using the robustness objective and one is trained without robustness objective. We modify CW attack to find attacks of confidence >= 0.9. On the first 30 random samples, only 2 confident attacks are found on the robust model, but 29 confident attacks are found on the natural model! This gives evidence that Madry et al.'s robust model is good.

3. (R3) Whether this paper assumes common classifiers are good, or is about a way to make classifiers good?

Please refer to point 1: Neither is the goal of this this paper. Existing common classifiers are unlikely to be good, and we seek through literature and decide to use Madry et al.'s model. We apologize for the confusion.

4. (R3) Weak connection with manifold assumption and Basri-Jacobs.

First, manifold assumption is critical for us to propose considering confidence information and the goodness property. The low dimensionality in manifold assumption is important to our intuition that handling low-confidence points are more challenging than separating confident regions.

Second, manifold assumption now manifests in our analysis. In Definition 5, if the data generating distribution has entangled classes, then generalization and robustness are in contradiction: If model generalizes well with correct and confident predictions, then in a small neighborhood there are two points which are predicted with different labels and each with high confidence. This implies poor separation and thus poor robustness.

Finally, there is a factual misunderstanding about Basri-Jacobs.  Basri-Jacobs do not only study points that fall on the manifolds, but also they study those nearby the manifolds. This is mentioned in their abstract, and the entire Section 4 is about this. Their work directly inspires our thought experiment and defenses.

5. (R3) CW attacks are about minimal perturbations that change labels, so they only produce low-confidence attacks.

First, existing attacks (including FGSM by Goodfellow et al.) all concern about changing labels. CW attacks are by far the strongest known attacks and subsume previous proposals.

Second, models used in Goodfellow et al. are weak (much smaller than Madry et al.'s and is not trained with robustness). Thus they are far from being good, and so FGSM found confident attacks with minimal perturbations. Our results indicate that it is much harder to generate high-confidence attacks on Madry et al.'s model.

Finally, the fact that CW attacks so far only find low-confidence attacks on Madry et al.'s model, in contrast to that FGSM finds high-confidence attacks on weak models, gives further evidence that Madry et al.'s model is good.

6. (R1) How about reinforcement learning?

We believe that the same underlying principle applies: One needs to take into account the confidence information of a model when dealing with adversarial perturbations. Specifically, since reinforcement learning is modeled as a Markov decision process, the confidence information naturally exists there (yet no work has taken it into account in dealing with adversarial noise!). We will give more details in a separate reply.

---

### Public Comment · (anonymous) · 2017-12-28
**A dataset for testing the manifold assumption**

This paper may be relevant to the current discussion: https://openreview.net/forum?id=SyUkxxZ0b.
It explores adversarial examples on a synthetic dataset where the data manifold is defined mathematically (classifying between two concentric spheres). For this dataset it is mathematically defined what is on and off the data manifold (just based on p(x)). The authors can find local adversarial errors both on and off the data manifold, the search space of a small L_2 ball will contain both kinds. In fact the average distance to the nearest "on the data manifold" and "off the data manifold" adversarial examples are comparable.

The basic conclusion of this paper is that, at least for this dataset, the reason local errors exist near most correctly classified data points is due to the geometry of the manifold itself, and not some other issue of the neural network classifiers. In fact, the authors can prove a bound relating the amount of test error to the average distance to nearest error, which implies that any model with non-zero test error must have local adversarial errors. It might be useful for the authors to discuss their defense proposal in the context of the sphere dataset given the simplicity of this dataset. It could help clarify what property of either the dataset, or the models the authors are hoping to capitalize on.

---

> ### Author Response · Authors · 2017-12-30
> **Quick thoughts**
>
> Thanks for the pointer. We will consider adding discussions and experiments based on this synthetic data set in future drafts. Some preliminary thoughts here:
>
> 1. Our method is not supposed to work with an arbitrary base model, but only good ones where "confident regions" are well separated. Taking this view, the thing is that you can use confidence to distinguish good and bad points. For example, even though average distance to the nearest "on manifold data" and "off manifold data" are comparable, we can still distinguish them because "off/far-away from manifold data" because good models have low confidence on them.
>
> 2. Perhaps most surprisingly, we find that good models already follow from existing training objectives. In particular, Madry et al.'s recent robust model already encourage good separation for confident points, but the control for low-confidence points is weak. One can prove such facts using tools as simple as Markov's inequality.
>
> 3. That said, we think it is good to try robust objective on the synthetic dataset and see how well confidence really works (note that we also need a good model architecture in order to fit well the geometric structure).

---

### Author Response · Authors · 2018-01-03
**Second response: Paper revision**

We have submitted a revision of our draft. Three most important changes:

1. Theoretically, we can now prove that Madry et al.'s objective encourages well-separated confident regions. Especially, predictions of different classes with confidence p will be well separated as soon as p increases. See Proposition 1 in Section 4 of the new draft. Moreover, our analysis also reveals that Madry et al.'s objective may have a weak control of low-confidence but wrong predictions in the sense that there might be many such points (see Proposition 2 in Section 4 of the new draft and discussions thereafter).

This new theoretical connection/evidence with Madry et al.'s paper is surprising to us, because it exactly matches the intuitions we developed from the manifold assumption.

2. Empirically, we have conducted a new experiment (due to time constraint this is the fastest experiment we can finish before revision deadline) which compares Madry et al.'s robust model with the natural model (which has the same architecture but is trained without any robustness objective) in terms of the separation of confident regions. The results are encouraging and conform to our theoretical analysis very well. In short, it is much harder to find confident attacks on the robust model than the natural model (where confident attacks almost always exist). Please see paragraph "Comparing models using the goodness property" in Section 6.

3. We have restructured the paper a bit so as to decouple "separation of confident regions" from "being confident on natural manifolds." (thus by manifold assumption confident regions must be separated). This decoupling makes our goal more clear (constructing a robust model requires consideration of separating confident regions, and embedding can be useful in fixing low-confidence errors), with the caveat that being able to generalize requires a learner to learn the manifolds.

---

### Decision · Program_Chairs · 2018-01-29
**ICLR 2018 Conference Acceptance Decision**

**Decision:**

Reject

**Comment:**

The original paper was sloppy in its use of mathematical constructs such as manifolds, made assumptions that are poorly motivated (see review #2 for details), and presented an empirical evaluation is preliminary. Based on the reviews, the authors have substantially revised the paper to try and address those issues by adding new theory, etc.

Unfortunately, it is difficult to assess whether these revisions are sufficient to address the aforementioned issues without going through a second round of "full" review. I encourage the authors to use the reviewer comments to further improve the paper, and re-submit to a different venue.